# Inducing Precision in Lagrangian Neural Networks: Proof of concept application on Chaotic systems

## Abstract

Solutions of dynamic systems that exhibit chaotic behavior are particularly sensitive to errors in initial/intermediate state estimates when long term dynamics is of interest. Lagrangian Neural Networks (LNN) are a class of physics induced learning methods that seamlessly integrate physical conservation laws into functional solutions, by forming a parametric Lagrangian for the system of interest. However it has been seen that the function approximation error associated with the parametric Lagrangian modelling could prove to be catastrophic for the prediction of long term dynamics of chaotic systems. This makes improving the precision of the parametric Lagrangian particularly crucial. Considering the same in this work a modified LNN approach is proposed, where a customized neural network architecture is designed to directly emphasize the relative importance of each significant bit in the Lagrangian estimates produced. We evaluate our method on two dynamic systems that are well known in the literature in exhibiting deterministic chaos, namely the double pendulum and Henon-Helies systems. Further, we compare the obtained solutions with those estimated by Finite Element solvers (under optimal conditions) to validate the relative accuracy. We observe that the trajectory deviations as a result of chaotic behavior can be significantly reduced by the process of explicitly enforcing the precision requirement for the parametric Lagrangian, as modelled using the proposed approach.

## 1 Introduction

Differential Equations (DEs) play a prominent role in the dynamic modelling of virtually every physical system Braun & Golubitsky (1983); Bandrauk et al. (2007). Accounting for the same, the literature associated with the solution estimation of differential equations are well versed. The simplest of which are solution methods involving the estimation of direct symbolic functional forms, note that the same have been developed only for a minor set of differential equation classes Bluman & Anco (2008). For the large majority of DEs which are used in the modelling of systems of practical interest, the closed form symbolic functional solutions cannot be estimated. A prominent workaround involves the use of Finite Element methods Logg et al. (2012), the same estimates point-wise numerical solutions of a given differential equation (for a set of points defined by a mesh-geometry) and interpolates the same to obtain the solution in a predetermined domain. Physics informed neural networks (PINNs) are a class of mesh-free alternative that provide functional solutions for DEs. A most recent study Grossmann et al. (2023) found that PINNs show lower computational cost (2 orders of magnitude) in comparison to Finite Element methods when estimating solutions of DEs of higher dimensionality, as the latter's computational complexity scales with the dimensionalilty due to the increased complexity of mesh required. PINNs being inherently mesh-free accounts for minimal domain points for solution estimation. PINNs could also integrate system configurations into the solution space Raissi et al. (2019), eliminating the need for re-estimation of the solution space in accordance with change of system parameters. Another notable advantage of PINNs is that they can seamlessly integrate domain information in the form of data collected from real world settings Raissi et al. (2017) as a set of additional constraints on the solution space. The typical set of DEs that could be solved using PINNs range from (but are not limited to) integer-order Raissi et al. (2019), integro-differential equations Sirignano & Spiliopoulos (2018); Lu et al. (2021), fractional

PDEs Pang et al. (2019), high dimensional-PDEs Zeng et al. (2022) or even stochastic PDEs Zhang et al. (2020); Yang et al. (2020). The applicability of the standard PINN based procedure carried out for estimating functional solutions for a given ordinary differential equation is well studied within multiple domains of science and engineering Bararnia & Esmaeilpour (2022); Misyris et al. (2020); Cai et al. (2021); Wang et al. (2022). In most PINN settings aimed to solve a governing differential equation of system (with state $x$), the functional solution $f(x)$ is approximated by numerical optimization of the neural network parameters Sexton et al. (1998) with the optimization objective as the residue (Equation 1) obtained from substituting the PINN functional solution and it's derivatives into the differential equation of interest.

$$^{res}\mathcal{O}_D = \frac{1}{n} \sum_{i=0}^{n} \left\| D_E \left( f(x_i),\ f^{'}(x_i),\ f^{''}(x_i),\ ... \right) \right\|^2 \tag{1}$$

Note that the residue based objective values are computed at discrete state points within a limited domain from which coordinate points are sampled in batches at random, the same is inherently efficient as compared to Finite Element alternatives, eliminating the need for complex mesh geometries. Given any differential equation with state a set of state variables $x$, the same can be written in the residual form : $D_E[f(x)] = 0$. The residues can be evaluated by considering a discretized domain over the state variables with $(n)$ sample state coordinates (dataset of input variables), usually obtained through uniform sampling over state variable bounds (domain). Note that another significant component is the boundary/initial conditions (BCs) required to form the unique solution of the corresponding differential equation. For most standard cases of PINN solution estimation the same could be added directly into the functional solution form, by re-parameterizing the output space of the PINN model. In the case of a re-parameterized solution $^p f(x)$, the overall objective loss uses in the optimization procedure of the PINN model could be directly estimated by substituting the same in Equation 1, where the corresponding functional or derivative components required to solve the differential equation at a given point could be computed using the re-parameterized solution and its corresponding derivatives (which could be computed using automatic differentiation procedure Baydin et al. (2018)). In its most simple form, the PINN based algorithm for the solution estimation of a differential equation (DE) is as shown in Algorithm 3. Note that the PINN based functional modelling as illustrated in Algorithm 3 has been proven to be applicable to many physical systems, provided that the given set of boundary conditions is sufficient enough to guarantee a unique solution for the governing equations to be solved. For most cases the same is the only essential condition for convergence (obtain optimal solution). The gradients (partial derivatives with respect to functional model parameters) necessary for the back-propagation step (parameter optimization) is computed using the automatic differentiation method for neural networks. Generally a typical stochastic gradient descent (or its derivatives) based optimization strategy Bottou (2012); Kingma & Ba (2014) is used to optimize the network weights. Specifically for solving differential equations using PINNs, the solution as provided by the PINN can be augmented to satisfy exactly the required initial conditions Lagaris et al. (1998), Dirichlet boundary conditions Sheng & Yang (2021), Neumann boundary conditions McFall & Mahan (2009); Beidokhti & Malek (2009), periodic boundary conditions Zhang et al. (2020); Dong & Ni (2021), Robin boundary and interface conditions Lagari et al. (2020). Since the augmentation so performed is inherently functional, the automatic differentiation procedure can still be followed for all necessary gradient computation operations. Note that the modelling brought in by re-parameterising the solutions so as to inherently satisfy the boundary conditions are largely efficient as compared to vanilla counterparts such as regularization terms in optimization objectives; a straightforward argument for the same can be made by analysing the complexity driven convergence trends of such a weighted objective function. In addition to the same, if some features of the DE solutions are known a priori, it is also possible to encode them in network architectures, for example, multi-scale features, even/odd symmetries of energy conservation Mattheakis et al. (2019), high frequencies Cai et al. (2020) and so on.

Energy conservation is of prime interest in many dynamical systems, which helps in a more robust modelling of long term dynamics. A classical implementation of the same is by explicit energy conservation formulations such as gradient based optimization of a weighted objective, in which a component of the objective function is an explicit comparison between the known total energy of a system with the energy estimate obtained from predicted quantities. However most of these approaches fail to generalise or achieve sufficient precision requirements, a prominent reason for the same would be due to the increase in the variance of the overall objective function used. A much

noted alternative that efficiently addresses this problem is the Hamiltonian Neural Network Grey-danus et al. (2019) (HNN) formulation where instead of a direct modeling of a solutions or explicit objective components for energy conservation, the Hamiltonian of a system is modelled, from which quantities of interest that define the complete system dynamics (usually position, momenta) can be easily derived. Given a set of generalized coordinates $\{\mathbf{q}, \mathbf{p}\}$ for a system with $N$ degrees of freedom, that is $\mathbf{q} = \{q_1, q_2, ..., q_N\}$ and $\mathbf{p} = \{p_1, p_2, ..., p_N\}$. The pair $(\mathbf{q}, \mathbf{p})$ represents the complete state of the system (usually positions and momenta). If the total energy of the of the system is conserved, the following (Equation 2) can be written in terms of the Hamiltonian analogue $\mathcal{H}$.

$$\frac{\partial \mathbf{q}}{\partial t} = \frac{\partial \mathcal{H}}{\partial \mathbf{p}} \quad , \quad \frac{\partial \mathbf{p}}{\partial t} = -\frac{\partial \mathcal{H}}{\partial \mathbf{q}} \tag{2}$$

Note that the left hand side of both equations (in Equation 2) is the constraint used to optimize the network parameters. In a data driven approach, the same maybe estimated using finite difference in time where the state evolution, i.e., data required is known. The complete objective function used for optimizing the Hamiltonian neural network is as shown in Equation 3.

$$\mathcal{O}_{HNN} = \left| \frac{\partial \mathbf{q}}{\partial t} - \frac{\partial \mathcal{H}}{\partial \mathbf{p}} \right|^2 + \left| \frac{\partial \mathbf{p}}{\partial t} + \frac{\partial \mathcal{H}}{\partial \mathbf{q}} \right|^2 \tag{3}$$

Similarly Lagrangian neural networks (LNNs) Cranmer et al. (2020) make use of observed states of a system to model the Lagrangian directly. The advantage of the same over HNNs comes from the ability of the Lagrangian to be defined over an arbitrary coordinates in space as opposed to strictly canonical spaces in the case HNNs. Given a set of coordinates $x_t = (q, \dot{q})$ we can make use of the Lagrangian as modelled using LNNs to estimate $\ddot{q}$ by making use of the modified Lagrangian equation Hand & Finch (1998) as follows,

$$\ddot{q} = \left( \frac{\partial^2 \mathcal{L}}{\partial \dot{q}^2} \right)^{-1} \left( \frac{\partial L}{\partial q} - \dot{q} \frac{\partial^2 \mathcal{L}}{\partial q \partial \dot{q}} \right) \tag{4}$$

By integrating Equation 4 after substituting the black-box Lagrangian $\mathcal{L}$, one could re-estimate the state variables $(q, \dot{q})$ as functions in time, solved for a specific set of initial condition. The re-estimated state variables could therefore by compared with pre known state variable data (similar to the HNN objective) to optimize the LNN. As per Grossmann et al. (2023) the most prominent limitations of differential equation solutions as estimated using vanilla PINNs is the larger relative error margins in the functional solutions as compared to Finite element alternatives. The time independent nature of solutions as estimated by LNNs makes it inherently superior to traditional alternatives, however the function approximation error in the Neural Network based solution is inherently higher than the Finite Element alternatives. Note that the same acts as a significant limitation to the LNN or HNN approach as a reduced precision in the Lagrangian or Hamiltonian parametric model causes significant solution deviations in systems that exhibit chaotic behavior Cranmer et al. (2020). In this work we hope to effectively address this limitation by adding explicit modelling constraints to preserve the numerical precision of the LNN model upto a specified limit.

## 2 METHODOLOGY

As per the Lagrangian Neural Network (LNN) formulation as proposed in Ref Cranmer et al. (2020), a functional (parametric) Lagrangian could be estimated for any system whose state $(x_t)$ at time $(t)$ is defined by a set of position and velocity coordinates $(\mathbf{q}, \dot{\mathbf{q}})$. Here the data $\mathcal{D}$ used consists of valid trajectory estimates $\mathcal{D} = \{q_n, \dot{q}_n\}$. In other words, the dataset $\mathcal{D}$ consists of $N$ randomly sampled trajectory points (initial values) sampled over the space of possible state variable $(\mathbf{q}, \dot{\mathbf{q}})$ values. By doing so one hopes to exhaust a wide range of possible configurations of the system of interest. The vectorized Lagrangian formulation of any dynamic system can be written as,

$$\frac{d}{dt} \nabla_{\dot{q}} \mathcal{L} = \nabla_q \mathcal{L} \tag{5}$$

Expanding the Equation 5 by propagating the time derivative through the gradient of the Lagrangian and isolating $\ddot{q}$ would result in Equation 6. The LNN models a parametric Lagrangian $\mathcal{L}$, which is

the direct predicted quantity of the neural network. Equation 6 can be applied on $\mathcal{L}$ so as to obtain an estimated functional form for $\ddot{q}_{pred}$, valid for a given domain and system specifics/configuration.

$$\ddot{q} = \left(\nabla_{\dot{q}}\nabla_{\dot{q}}^\top \mathcal{L}\right)^{-1} \left[\, \nabla_q \mathcal{L} - (\nabla_q \nabla_{\dot{q}}^\top \mathcal{L})\dot{q} \,\right] \tag{6}$$

Note that trajectory of the state variables can be estimated through numerical integration of $\ddot{q}_{pred}$ obtained using Equation 6. Now one can easily form the basis for the predictive error used for optimizing the parametric Lagrangian $\mathcal{L}$ as modelled by the Lagrangian Neural Network. That is the optimization constraint is to minimize the predictive error between the trajectory estimates obtained by integrating the estimated $\ddot{q}_{pred}$ and known values of state variables $(\mathbf{q},\ \dot{\mathbf{q}})$.

## 2.1 MOTIVATION

This work builds upon the LNN formulation Cranmer et al. (2020), in an effort to effectively address the function approximation error associated with the modelling of the parametric Lagrangian $\mathcal{L}$. We show that by adding additional constraints aimed at the numerical precision of the Lagrangian estimates, the deviations that could occur while estimating long term solutions could be considerably addressed. That is, as a direct consequence of a more precise Lagrangian estimate obtained, the latter estimates of quantities (state variables) that emerge from same are also inherently better in terms of precision. In a mathematical perspective, the precision improved Lagrangian directly results in a significantly lower cumulative integration error, when estimating the trajectory states $(q,\ \dot{q})$ by numerical integration of $\ddot{q}_{pred}$. This is particularly important for chaotic systems where incurring numerical errors during the solution estimation process can cause significant changes in the long term estimates of solutions or trajectories Duit et al. (1998). In this work we introduce a scalable formulation that can explicitly and independently condition on the accuracy of each significant digit in the solution space obtained, when using a Lagrangian Neural Network.

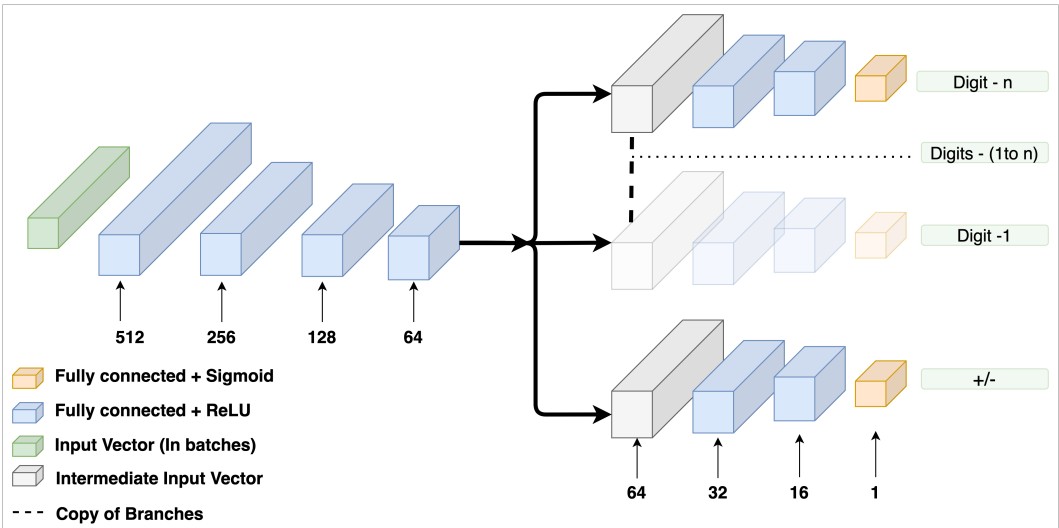

Figure 1: Branched Multi-Output Deep Neural Network architecture used in this study, where each output branch is used to independently predict each significant digit of the output quantity.

## 2.2 PRECISION CONSTRAINT

Suppose the dataset $\mathcal{D}$ used to train the LNN consists of $N$ sample points (initial values) and let the predicted state estimate at $t = 0$ obtained by integrating the $\ddot{q}_{pred}$ (obtained from plugging in the LNN into Eqn. 6) be $(q_{\text{pred}},\ \dot{q}_{\text{pred}})$, then the objective loss used for the optimization of the LNN is the average of $L_2$ norm values (Mean Squared Error) over all residues (each data-point) as shown in Eqn. 7.

$$\mathcal{O}_{pred} = \frac{1}{N} \sum_{i=1}^{N} \|q_{pred}^i - q_{data}^i\|^2 + \|\dot{q}_{pred}^i - \dot{q}_{data}^i\|^2 \tag{7}$$

The proposed advancement in this work can be split into two parts: 1. The modified neural network architecture, and 2. The regularization terms added into the overall objective function (loss) during the parametric optimization process (training) of the LNN. The deep neural network used in this study follows a doubling sequence neural scheme by which the number of layers of the same could be scaled to arbitrary values. Also note that the neural network used is infinitely differentiable (end to end), i.e., the partial derivatives of any order with respect to any input quantity is directly estimated using automatic graph differentiation Griewank et al. (1989). The typical structure of the neural network architecture used involves predicting each digit of the output quantity ($\mathcal{L}$) using a separate branch, the complete architecture is as illustrated in Fig. 1. Note that to effectively use this method we first fix the required number of digits that are of interest based on the overall output range. For simplicity we assume that the predicted Lagrangian estimate is the $n$-bit binary integer representation of the actual Lagrangian. Sigmoid activation is used to bound and convert the predicted value (output) of each branch to directly estimate the corresponding bit of the binary signed integer representation of the quantity being predicted. Equation 8 is used to recover the value of the Lagrangian post the first stage of model output. Where, $B_0$ is the rounded output value (rounded value of sigmoid output) of the branch that predicts the sign of the integer, and each $B_i$ correspond to the rounded output of the branches that predicts the bit value corresponding to each binary digit. Also note that $k$ is a parameter that directly decides the number of decimal places in the model output post conversion.

$$\mathcal{L}_{\text{pred}} = \frac{1}{10^k} B_0 \sum_{i=1}^{n} 2^i \times B_i \tag{8}$$

The aim of the added regularization term in the objective function is to ensure that the prediction of each binary bit is accurate once the parametric optimization process of the neural network is complete. The expected dataset to train the LNN using the added regularization factor is a set of initial conditions ( state variables : $\{q_i, \dot{q}_i\}$ ) and Analytical Lagrangian ($\mathcal{L}$) values, that are randomly sampled over a range of possible configurations of state variables. Let us suppose that the dataset consists of $N$ data-points, each with true Lagrangian estimates $\mathcal{L}^i$ calculated using the corresponding Lagrangian equation of the system of interest. Further, let $B_j^{\mathcal{L}^i}$ be the $j^{\text{th}}$ binary value in the signed binary representation of $\mathcal{L}^i$ post scaling the same using the factor $10^k$. Then the corresponding regularization factor to be added to the overall objective loss of the LNN is estimated as the binary cross entropy value between $B_j^{\mathcal{L}^i}$ and the predicted sigmoid probabilities $B_j^p$ for each corresponding bit as shown in Eqn. 9.

$$\mathcal{O}_{\text{reg}} = \frac{-1}{N} \sum_{i=1}^{N} \sum_{j=0}^{n} \left[ B_j^{\mathcal{L}^i} \times \log(B_j^p) - (1 - B_j^{\mathcal{L}^i}) \times \log(1 - B_j^p) \right] \tag{9}$$

Now, the total objective loss used in the parametric optimization of the modified LNN is the weighted sum of the prediction loss (Eqn. 7) and the regularization factor (Eqn. 9) as shown in Eqn. 10, where $\lambda$ is the scale factor.

$$\mathcal{O}_{\text{total}} = \mathcal{O}_{\text{pred}} + \lambda \mathcal{O}_{\text{reg}} \tag{10}$$

Although the analytical form of the Lagrangian is know for a large set of physical systems, it's to be noted that a significant limitation of this approach is the requirement of the same. The proceeding section explores a notable alternative approach to form intermediate temporal estimates of the Lagrangian in hope of eliminating the need for an analytical Lagrangian for the system of interest.

### 2.3 STEADY STATE ASSUMPTION - PROPOSAL

In this section we introduce a regularization strategy to enforce precision using the formulation mentioned in the preceding section when the explicit Lagrangian equation of the system of interest

in not known. We base the regularization on the fact that the output estimate of the LNN at epoch $(e-1)$ in the model training process should be equivalent to the estimate at epoch $(e)$ when the modelling is close to being optimal, as the quantity being approximated has no associated stochastic nature. Let us suppose ${}^e\mathcal{L}_{\text{pred}}$ is the predicted output by the LNN, where $e$ denote the epoch count of the LNN training process, the Lagrangian estimate at epoch $e-1$ (just preceding) could be used as the pseudo true estimate for estimating the precision regularization term (Eqn. 9) for the estimate at the current epoch $(e)$, acting as an alternative to the true value. The precision based regularization term can now be estimated using Eqn. 9 considering $\mathcal{L}_{\text{pseudo}}^i$ as the true estimate. Additionally it's necessary to enforce that the temporal error on the Lagrangian estimates given by the LNN is minimized by training progress. This is particularly clucial to ensure the validity assumption on the invariant nature of the estimates formed under optimal model conditions. We estimate the temporal convergence regularization term as shown in Eqn. 11, when the training dataset $\mathcal{D}$ contains $N$ sample points.

$$\mathcal{O}_{\text{TC}} = \frac{1}{N} \sum_{i=1}^{N} \left\| {}^e\mathcal{L}_{\text{pred}}^i - {}^{e-1}\mathcal{L}_{\text{pred}}^i \right\|^2 \tag{11}$$

Note that in most cases the scale of the quantity being approximated is unknown apriori. Since stability of the numerical value of the Lagrangian should not depend on the relative scale variations of the same between sample points, we follow a normalization procedure as outlined in Algorithm 1 for the purposes of estimating Eqn. 11 invariant of the relative scale of each Lagrangian estimate $(\mathcal{L}_{\text{pred}}^i)$.

---

**Algorithm 1:** Intermediate Estimate for Temporal Convergence Assessment

**Data:** No of significant digits : $n$, Model Prediction : $\mathcal{L}_{pred}^i$
**Result:** Obtain : ${}^{temp}\mathcal{L}_{pred}^i$
**Estimate Current Scale for Normalization**
$L_n = log_{10}(\left|\mathcal{L}_{pred}^i\right|)$;
$L_n = \lfloor L_n \rfloor$
**Normalize the Output Using Scale Estimate**
${}^{norm}\mathcal{L}_{pred}^i = \mathcal{L}_{pred}^i \times 10^{-L_n}$;
**Upscale to Meet Precision Requirements**
${}^{temp}\mathcal{L}_{pred}^i = {}^{norm}\mathcal{L}_{pred}^i \times 10^n$
**Return** ${}^{temp}\mathcal{L}_{pred}^i$

---

Another factor to note is that the output values produced in the initial stages of neural optimization procedures are largely transient, enforcing temporal convergence regularization to stabilize the model predictions at such stages causes sub-optimal convergence. To ensure generality of the overall algorithm we weight the temporal convergence loss term ($\mathcal{O}_{\text{TC}}$) with an training epoch dependent factor that increases (linear scaling) with training progress.

$$\mu_{TC} = \mathcal{G} \left( \frac{C_{epoch}}{N_{epochs}} \right) \tag{12}$$

A typical example of the scale factor used to relatively scale the temporal convergence component of the objective is as shown in Equation 12. Where, $C_{\text{epoch}}$ is the current epoch and $\mathcal{G}$ is a tunable scalar hyper-parameter, that decides the overall relative weight of the temporal convergence component to total objective loss of the LNN. The total objective loss for the LNN optimization can be written as the relative weighted sum of the three loss components ($\mathcal{O}_{\text{pred}}$, $\mathcal{O}_{\text{reg}}$, $\mathcal{O}_{\text{TC}}$) as shown in Eqn. 13.

$$\mathcal{O}_{total} = \mathcal{O}_{pred} + \mu_{TC} \times \mathcal{O}_{TC} + \lambda \times \mathcal{O}_{reg} \tag{13}$$

Also note that the presence of regularization terms in the objective function is directly compatible with any Auto-grad based stochastic gradient strategy used to optimize the weights of the neural network, since the expected value of the overall objective is still tending to zero in the asymptotic

---

**Algorithm 2:** LNN training process with implicit precision enforcement.

---

**Data:** Span of Initial States : $\mathcal{D} = \{q_i, \dot{q}_i\}$, Precision : $k$
**Result:** Lagrangian Neural Network : $\mathcal{L}_{NN}( q, \dot{q} )$
**Initialize LNN model with random weights**
Model $(\mathcal{L}_{NN})$ = Create-New-Neural-Function-Approxmiator();
**Iterate for no of epochs**
**for** $e$ = 1 to $N_{epochs}$ **do**
    **Estimate** $\mathcal{L}_{pred}$ **using current** $\mathcal{L}_{NN}$
    $^e\mathcal{L}_{pred}$ = Forward Pass $\mathcal{L}_{NN}( q, \dot{q} )$;
    **Estimate** $\ddot{q}$ **using current** $^e\mathcal{L}_{pred}$
    $\ddot{q}_{pred}$ = estimated using Equation 6 : $\mathcal{L} =^e \mathcal{L}_{pred}$ ;
    **Estimate State Variables :** ($q_{pred}$, $\dot{q}_{pred}$**) using** $\ddot{q}_{pred}$
    $q_{pred}$, $\dot{q}_{pred}$ = Perform Numerical Integration on : $\ddot{q}_{pred}$ ;
    **Estimate prediction Objective for state estimate**
    $\mathcal{O}_{pred}$ obtained using Equation 7;
    **Conditional to initialize** $^{e-1}\mathcal{L}_{pred}$ **estimate**
    **if** $e == 1$ **then**
        **Initialize the value of** $^{e-1}\mathcal{L}_{pred}$
        $^{e-1}\mathcal{L}_{pred} = {}^e\mathcal{L}_{pred}$;
        **Backpropagate prediction loss on** $\mathcal{L}_{NN}$ **weights :** $w$
        $w_j = w_j - \alpha\frac{\partial}{\partial w_j}\mathcal{O}_{pred}(w)$;
        **Skip next steps in loop**
        *continue*;
    **end**
    **Normalize predicted Lagrangian Estimates**
    $^{e-1}\mathcal{L}_{pred}$ = Using Algorithm 1 : $(^{e-1}\mathcal{L}_{pred}, k)$;
    $^e\mathcal{L}_{pred}$ = Using Algorithm 1 : $(^{e-1}\mathcal{L}_{pred}, k)$;
    **Calculate Temporal Convergence Objective**
    $\mathcal{O}_{TC}$ = Using Equation 11 : $(^{e-1}\mathcal{L}_{pred}, {}^e\mathcal{L}_{pred})$;
    **Calculate Precision Regularization Objective**
    $\mathcal{O}_{reg}$ = Using Equation 9 : $(^{e-1}\mathcal{L}_{pred}, {}^e\mathcal{L}_{pred})$;
    **Calculate Total Objective Value**
    $\mathcal{O}_{total}$ = Using Equation 13;
    **Backpropagate total loss on** $\mathcal{L}_{NN}$ **weights :** $w$
    $w_j = w_j - \alpha\frac{\partial}{\partial w_j}\mathcal{O}_{total}(w)$;
    **Reset the value of** $^{e-1}\mathcal{L}_{pred}$
    $^{e-1}\mathcal{L}_{pred} = {}^e\mathcal{L}_{pred}$;
**end**
**Return final converged** $\mathcal{L}_{NN}( q, \dot{q} )$

---

case. However the local behaviors of convergences such as rate of convergence and variance in objective evolution can significantly change. Further theoretical analysis is required to gain more clarity regarding the same.

## 3 EXPERIMENTS AND RESULTS

The experiments are carried out as a comparative analysis over the proposed setting and existing alternatives, the analysis is conducted based on the relative error margins. Order four Runge-Kutta method Tan & Chen (2012) with a tolerance set to $10^{-13}$ was used to estimate the acceptable ground truth (*Exact* solution) for all systems under consideration. We also evaluate the trajectory deviations of the best model under long timescale solution estimations for relative comparison of model settings. Training and testing code for all models and settings were developed using the python JAX framework Schoenholz & Cubuk (2020). All experiments were conducted using an 8 core 9th gen intel i7 processor and a Tesla P4 GPU.

### 3.1 DOUBLE PENDULUM

The ability to estimate trajectories with minimal deviations in systems under chaotic behaviour is of prime interest for this study, hence, all experiments are conducted on two classical systems that are proven to exhibit chaotic behaviour. Firstly we employ the present methodology to approximate solutions to a classical system that is proven to have deterministic chaos behavior, which is double pendulum. The dynamic equations of the double pendulum used here are as derived in Ref Assencio.

$$\alpha_1 = \frac{l_2}{l_1}\left(\frac{m_2}{m_1 + m_2}\right)\cos(\theta_1 - \theta_2), \quad \alpha_2 = \frac{l_1}{l_2}\cos(\theta_1 - \theta_2) \tag{14}$$

$$f_1 = -\frac{l_2}{l_1}\left(\frac{m_2}{m_1 + m_2}\right)\dot{\theta_2}^2\sin(\theta_1-\theta_2)-\frac{g}{l_1}\sin(\theta_1), \quad f_2 = \frac{l_1}{l_2}\dot{\theta_1}^2\sin(\theta_1-\theta_2)-\frac{g}{l_2}\sin(\theta_2) \tag{15}$$

To write the governing equations of the double pendulum system in a simplified manner, the following quantities $\{\alpha_1, \alpha_2, f_1, f_2\}$ are introduced as shown in Equations 14 and 15. The equations of motion of the double pendulum could now be written as shown in Equation 16.

$$\ddot{\theta}_1 = \frac{f_1 - \alpha_1 f_2}{1 - \alpha_1\alpha_2}, \quad \ddot{\theta}_2 = \frac{f_2 - \alpha_2 f_1}{1 - \alpha_1\alpha_2} \tag{16}$$

Note that the masses $\{m_1, m_2\}$ and the lengths $\{l_1, l_2\}$ decide the configuration of the system, for all experimental settings we set them equal to unity and the acceleration due to gravity $(g)$ is assumed to be $9.8\ m/s^2$. The Dataset $(\mathcal{D})$ used to train the LNN on the given double pendulum system consists of $6 \times 10^5$ randomly sampled initial states in the format $\{q^i = (\theta_1^i, \theta_2^i), \dot{q}^i = (\dot{\theta_1}^i, \dot{\theta_2}^i)\}$ and corresponding Lagrangian values here are estimated using Equation 17.

$$\mathcal{L} = \frac{1}{2}(m_1 + m_2)l_1^2\dot{\theta}_1^2 + \frac{1}{2}m_2\dot{\theta}_2^2 + m_2 l_1 l_2 \dot{\theta}_1 \dot{\theta}_2 \cos(\theta_1 - \theta_2) + (m_1 + m_2)gl_1\cos\theta_1 + m_2 gl_2\cos\theta_2 \tag{17}$$

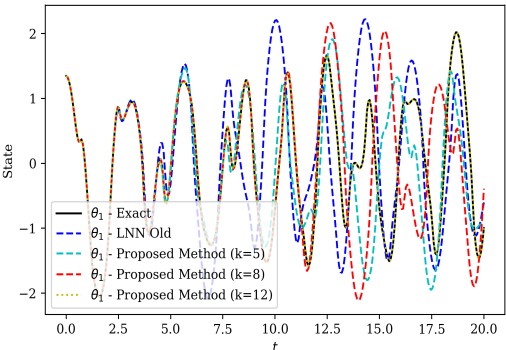

(a) Comparison of trajectory estimates obtained using numerical and LNN based methods.

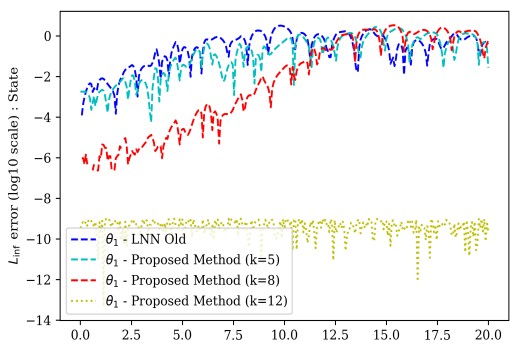

(b) Absolute error comparison between LNN based solutions and numerical solver.

Figure 2: Comparative results obtained on the double pendulum system, starting from $t = 0$ and given initial state : $\{q = (\theta_1 = \frac{3\pi}{7}, \theta_2 = \frac{3\pi}{4}), \dot{q} = (\dot{\theta}_1 = 0, \dot{\theta}_2 = 0)\}$

### 3.2 HENON-HEILES

The second system considered for the evaluation on the applicability of the proposed method is the Henon-Heiles system Hénon & Heiles (1964), the same models the non-linear motion of a star

around a galactic center with the motion restricted to a plane. In it's simplest form the Equation of motion of the system can be described using Equation 18. Where, the system is described using the position $q = (x, \ y)$ and momentum coordinates $\dot{q} = (p_x = \dot{x}, \ p_y = \dot{y})$ of the star.

$$\dot{p_x} = -x - 2\lambda xy, \quad \dot{p_y} = -y - \lambda(x^2 - y^2) \tag{18}$$

Note that $\lambda$ is set to 1 for all experimental configurations of the Henon-Heiles system (widely applied in the chaos literature). The Dataset ($\mathcal{D}$) used to train the LNN on Henon-Heiles system consists of $6 \times 10^5$ randomly sampled initial states in the format $\{q^i = (x^i, \ y^i), \ \dot{q}^i = (p_x^i, \ p_y^i)\}$ and corresponding Lagrangian values can be estimated using Equation 19.

$$\mathcal{L} = \frac{1}{2}(p_x^2 + p_y^2) - \frac{1}{2}(x^2 + y^2) - x^2 y + \frac{1}{3}y^3 \tag{19}$$

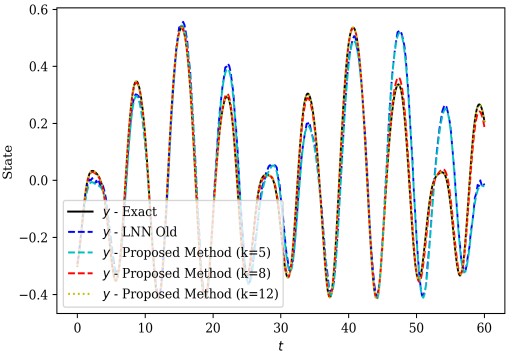

(a) Comparison of trajectory estimates obtained using numerical and LNN based methods.

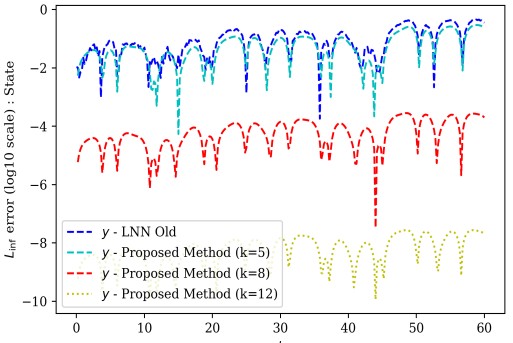

(b) Absolute error comparison between LNN based solutions and numerical solver.

Figure 3: Comparative results obtained on the Henon-Heiles system, starting from $t = 0$ and for the given initial state $\{q = (x = 0.3, \ y = -0.3), \ \dot{q} = (p_x = 0.3, \ p_y = 0.15)\}$

## 4 CONCLUSIONS

In this a work a new augmented form of the Lagrangian Neural Network is introduced which can extend the precision of the parametric Lagrangian to arbitrary extends. In our comparison on setting the precision ($k = 5$) of the proposed method close to what can be achieved by the original LNN formulation Cranmer et al. (2020), it can be seen that (Figures 2 and 3) the proposed method obtains comparable solutions and error margins as compared to the LNN approach Cranmer et al. (2020). Additionally we showed that the proposed method can scale the precision of the parametric Lagrangian to arbitrary numerical precision (at the cost of a larger parametric space) and effectively address the shortcomings associated with solution estimation of systems that exhibit deterministic chaos behavior. The feasibility of the proposed method to extend the achievable numerical precision due to computational limitations is discussed in Appendix C. Even-though there is a relative (but feasible) increase in computational resources required, the method is proven to help in maintaining the validity of the solution spaces of systems that exhibit chaotic behavior for longer time intervals. From relative comparison with the existing literature (Appendix D) it can be seen that the proposed approach preserves the arbitrary nature of required coordinates the LNN Cranmer et al. (2020) had, with an addition of being able to preserve arbitrary precision standards on the solution space formed. In future work we hope to evaluate the applicability of the same in chaotic systems with considerably larger degrees of freedom and also extend the Lagrangian formulation so as to directly include arbitrary configurations (instead of single) of physical systems in the parametric Lagrangain formed.

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

## A    VANILLA PHYSICS INFORMED NEURAL NETWORKS

Algorithm 3 roughly outlines the original physics informed neural networks based solution estimation strategyRaissi et al. (2017) used to estimate solutions of differential equations for a given boundary/initial condition.

---

**Algorithm 3:** Solution Estimation of Differential Equations using PINN

---

**Data:** Domain : $[a, b]$, Differential Equation : $D_E(f) = 0$, Sufficient $BCs$
**Result:** $^p f(x) = $ Re-parameterise $(\mathcal{F}_{NN}(x),\ BCs)$
**Initialize model with random weights**
Model $(\mathcal{F}_{NN})$ = Create-New-Neural-Function-Approxmiator();
**Iterate for a set number of epochs**
**for** $i = 1$ to $N_{epochs}$ **do**
    **Predict solution $^p f(x)$ by re-parameterizing current estimate**
    $^p f_{pred}(x) = $ Re-parameterise $(\mathcal{F}_{NN}(x),\ BCs)$;
    **Estimate necessary gradients using automatic differentiation**
    $^p f'_{pred}(x), \, ^p f''_{pred}(x), ...$;
    **Estimate prediction loss for current solution estimate**
    $\mathcal{L}_D$ - obtained using Equation 1 within the domain;
    **Backpropagate prediction loss**
    $w_{j+1} = w_j - \alpha \frac{\partial}{\partial w_j} \mathcal{O}_D(w)$;
**end**
**Return final converged solution $^p f(x)$**

---

It's worth noting the solution estimated using Algorithm3 is only valid for the limited Domain : $[a, b]$ and a given boundary/initial condition(s) $BCs$. Although the method effectively addressed the curse of dimensionalityKöppen (2000) as shown by finite element solvers, the limited validity of the solution space is considered as a backdrop which Lagrangian Neural NetworksCranmer et al. (2020) effectively address.

## B    HYPERPARAMETER SETTING USED

Additionally we outline the simplified network architecture tuning mechanism used in this study Figure 4. The network architecture definition is simplified into three hyperparameters : Minimum number of Neurons in the last hidden layer ($M_n$), Number of hidden layers ($M_n$) and Neuron factor (f). The exact value of these parameters are as mentioned in Table 1.

| Hyper Parameter | Value |
|---|---|
| No. Epochs | 100000 |
| Batch Size | 256 |
| Objective Loss | Equation 10 |
| Optimizer | Adam |
| Learning Rate | 0.005 |
| Adam - $\beta_1$ | 0.9 |
| Adam - $\beta_2$ | 0.999 |
| Adam - $\epsilon$ | 1e-08 |
| Network - $(M_n, M_n, f)$ | 16, 5, 2 |

Table 1: Hyper-parameter setting used in this study, all hyper-parameters were obtained by grid search. Experiment tracking done through wandb.aiBiewald (2020).
.

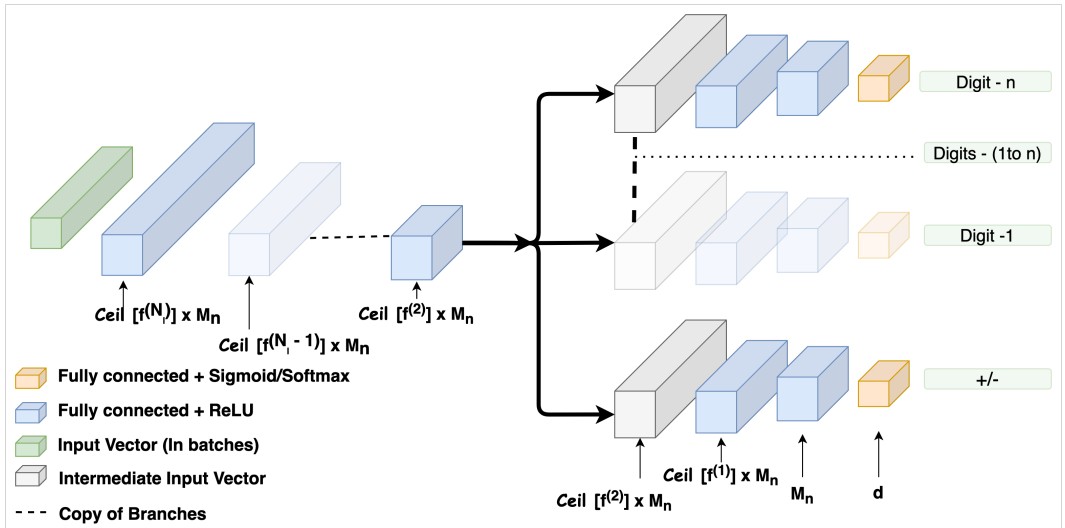

Figure 4: Illustration of the scheme by which network architecture tuning was carried out.

## C A NOTE ON NETWORK PARAMETERS

A crucial point of this study is the relative scalability of the introduced method. Although for our experiments we have used binary representation of integers as the underlying numeral system, the same is not unique and can be extended to numeral systems with larger integer bases. In which case the number of branches $n$ of the architecture used 1 could be substantially reduced.

Form Table 2 it can be seen that the number of trainable parameters for each branch (that predicts a binary bit) in the architecture 1 can be counted as $(2080 + 528 + 17 = 2625)$. An example scaling of the proposed method could to improve the number of bits, say by the addition of another 16 bits onto the current architecture. Note that the addition of $\tilde{4}2000$ parameters per each 16 binary bits is relevant although feasible computationally.

| Layer : (Branch-Type : No) | Output Shape | No Param |
|---|---|---|
| ReLU Linear : (Common : 1) | [batch-size, 512] | 2560 |
| ReLU Linear : (Common : 2) | [batch-size, 256] | 131,328 |
| ReLU Linear : (Common : 3) | [batch-size, 128] | 32,896 |
| ReLU Linear : (Common : 4) | [batch-size, 64] | 8,256 |
| ReLU Linear : (Branch : 1) x $n$ | [batch-size, 32] | 2080 x $n$ |
| ReLU Linear : (Branch : 2) x $n$ | [batch-size, 16] | 528 x $n$ |
| Sigmoid Linear : (Branch : 3) x $n$ | [batch-size, 1] | 17 x $n$ |

Trainable params (max): 301,040 ($n = 48$)
Non-trainable params: 0
Total params (max): 301,040

Table 2: Layer-wise parameter count of the Neural Network used under high precision binary setting.

# D    ADDITIONAL LITERATURE COMPARISON

The proposed work builds upon the Lagrangian Neural Network formulation (LNN) Cranmer et al. (2020), which enabled Lagrangian modelling of arbitrary dynamic systems represented in arbitrary coordinates systems. The LNN formulation improved the flexibility of the DeLaN Lutter et al. (2019) formulation by implicitly modelling the Lagrangian, improving the applicability of the same and better obeyed conservation laws. Finzi et al. (2020) augmented the LNN Cranmer et al. (2020) and Greydanus et al. (2019) formulations to work with Cartesian coordinates, which allowed the explicit addition of common physical constraints that better defines any dynamic system of interest, improving the data-efficiency and solution accuracy. Zhong et al. (2021) builds upon the Cartesian coordinate based formulations proposed in Finzi et al. (2020) to add explicit contact modelling constraints, which proved significant in improving the solution quality in dynamic systems for which contact forces were of crucial interest. In this work we take a detour from the Cartesian coordinate based formulations to directly imporve the LNN Cranmer et al. (2020) formulation by explicitly conditioning on the precision of the solution estimates. By doing so the proposed approach illustrates a scalable approach to achieve arbitrary precision standards, allowing for long term (time domain) validity of solutions to systems that exhibit deterministic chaos.

| | Neural Networks | Neural ODE | HNNs Greydanus et al. (2019) | DeLaN Lutter et al. (2019) | LNN Cranmer et al. (2020) | CHNN Finzi et al. (2020) | CLNN Finzi et al. (2020) | CM-CD-CHNN Zhong et al. (2021) | CM-CD-LNN Zhong et al. (2021) | This Work |
|---|---|---|---|---|---|---|---|---|---|---|
| Model Dynamic Systems | ✓ | ✓ | ✓ | ✓ | ✓ | ✓ | ✓ | ✓ | ✓ | ✓ |
| Differential Equation Solutions | | ✓ | ✓ | ✓ | ✓ | ✓ | ✓ | ✓ | ✓ | ✓ |
| Exact Conservation Laws | | | ✓ | ✓ | ✓ | ✓ | ✓ | ✓ | ✓ | ✓ |
| Learn from arbitrary coordinates | ✓ | ✓ | | ✓ | ✓ | | | | | ✓ |
| Learn from Cartesian Coordinates | ✓ | ✓ | | | | ✓ | ✓ | ✓ | ✓ | |
| Learn arbitrary Lagrangian | | | | | ✓ | | ✓ | | ✓ | ✓ |
| Explicit system constraints | | ✓ | | | | ✓ | ✓ | ✓ | ✓ | |
| Contact and Collision constraints | | ✓ | | | | | | ✓ | ✓ | |
| Adaptable to other works | | | | | | | | | | ✓ |
| Arbitrary precision constraints | | | | | | | | | | ✓ |
| Long term solutions: chaos | | | | | | | | | | ✓ |

Table 3: Outline of the existing literature and their relative strengths on comparision with the work proposed here.

