# OpenReview forum: "Inducing Precision in Lagrangian Neural Networks : Proof of concept application on Chaotic systems"
_ICLR.cc/2024/Conference — Submitted to ICLR 2024_

### Official Review · Reviewer_88d6 · 2023-10-26

**Soundness:** 3 good
**Presentation:** 3 good
**Contribution:** 2 fair
**Rating:** 5
**Confidence:** 3

**Summary:**

This paper aims to approximate chaotic systems using Lagrangian Neural Networks (LNN) with better precision. A new LNN architecture is proposed to emphasize the importance of significant bits. A new regularization term is added to ensure the accuracy of each significant bit. Experimental results demonstrate that the proposed LNN can achieve better precision.

**Strengths:**

1. It is important to pursue precision when approximating chaotic systems since a small error can cause long-term large errors.
2. The proposed method is succinct and easy to understand.
3. Experimental results verify the efficacy of the proposed method.

**Weaknesses:**

The motivation of the regularized term is not convincing. Adding a regularization term is a tradeoff between the original objective (Eq. (7)) and the regularization term (Eq. (9)). In common cases, minimizing the regularization term will make the original objective larger. In this paper, the original objective cares more about higher decimals, while the regularization treats all decimals equally. Thus, adding the regularization term will inevitably sacrifice the accuracy of higher decimals, which hurts the original objective. To my understanding, the original objective already reflects the precision requirement when approximating chaotic systems, and the regularization term only plays a negative role.

**Questions:**

There are many minor problems in this paper. Part of them are listed below.
1. In the tile, the colon is closer to proof rather than networks.
2. In the abstract, the abbreviation LNN appears on the 3rd line but its full name still occurs on the 10th line. After the abbreviation appears for the first time, it would be better to use the abbreviation rather than the full name.
3. On the 2nd line in the introduction, the citation is of the form "Name (year)". But the name is not a part of the sentence, thus it would be better to use the form "(Name, year)".
4. Above Eq. (1), "it's" should be "its".
5. The first paragraph in the introduction is too long (more than 1 page), and readers may lose the central idea easily. It would be better to separate it into several paragraphs.
6. Below Eq. (2), "maybe" should be "may be".
7. Significant digits or significant bits are not defined.
8. The form of citation of equations is not unified. Both "Equation 6" and "Eqn. 6" occur.

---

> ### Author Response · Authors · 2023-11-20
>
> The authors of this paper would like to thank the reviewer for the considerate comments and verbose and corrections pointed out. The revised manuscript has addressed the comments.
>
> Regarding Weaknesses pointed out :
>
> A straightforward explanation would be that the relative squared error approximates heavily on solution space approximates where there is considerable deviation from the expected solution space. In other words the squared error approximates has a natural weighting towards the most significant bits of interest, due to the relative increase of its penalty magnitude in the event of misprediction of a more significant bit. The regularization term acts as the explicit and equally weighted conditioning for estimating the relative accuracy of the predicted set of significant bits, independently. To sum up, the negative log likelihood loss must be minimized along with the prediction loss so as to obtain optimality.

---

> > ### Comment · Reviewer_88d6 · 2023-11-20
> >
> > Thanks for answering. It is true that both significant bits (i.e., higher decimals) and other bits (i.e., lower decimals) are important and we should optimize all of them in the ideal case. But in most problems, we cannot minimize the square loss term and the regularization term at the same time. By adding the regularizer, we may sacrifice the square loss. Thus, I persist that the regularization is not well motivated and prefer to maintain my rating unchanged.

---

> > > ### Author Response · Authors · 2023-11-21
> > >
> > > The authors concur with the fact stated by the reviewer, minimising an objective loss that contains the sum of a squared error and a negative log likelihood loss might not converge at all cases, mainly due to the relative scale variations of the two loss components. More specifically, with respect to the contributing gradient value of each component on a specific weight update, the loss component with the larger magnitude overshadows the lower, and MSE from prior literature is usually observed to be of considerably lower in magnitude as compared to Cross Entropy. However, in this work the same is specifically addressed by using a relatively weighted loss function (Equation. 10), where an explicit hyper-parameter (Lambda) is used to tackle the specific issue pointed out.

---

> > > > ### Comment · Reviewer_88d6 · 2023-11-22
> > > >
> > > > To my current understandings, the regularization is a kind of reweighting: significant bits have large weights in the square loss term, while all bits are treated in the same way in the regularization term. This means that the regularization term makes the significant bits less significant. Then minimizing the regularized objective will lead to more accurate insignificant bits and less accurate significant bits (compared to minimizing the square loss). As a result, I believe that less accurate significant bits (outputted by minimizing the regularized objective) will hurt the model more than less accurate insignificant bits (outputted by minimizing the square loss). Based on this, I am still not convinced of the proposed reguarization method.

---

> > > > > ### Author Response · Authors · 2023-11-22
> > > > >
> > > > > As added clarification, the objective for the LNN optimisation as proposed in this work (Equation. 10) consists of two terms
> > > > >
> > > > > the MSE component (derived from the Original LNN formulation) which acts as the relative weighted loss for the Lagrangian estimates produced. That is in the event of mis-prediction of the most significant bits of the Lagrangian estimate, the MSE component is significantly larger as compared to mis-prediction of the lower significant bits.
> > > > >
> > > > > the Regularisation component (part of the proposed approach) aims to solves the problem: every "bit" of interest no matter how low in-terms of relative magnitude contribution should be accurate, since the same is important in many settings that are considered solvable by Lagrangian Neural Networks. An example is when estimating solutions of systems that exhibit deterministic chaotic behaviour. The regularisation term thus provide a non-zero gradient contribution tell each bit of interest is predicted right. If the same is relatively weighted based on the relative significance of each bit, the shortcomings as exhibited by classical relative wighted objectives such as squared error would follow. Since the gradient contribution of the least significant bits would again be weighted down, and leave the optimisation of the same unaddressed.
> > > > >
> > > > > On added note, the objective functions are non-conflicting : that is the optimality of one does not negatively effect another, and potential differences in relative contributions are addressed by the relative weighting factor (lambda), as in Equation. 10.

---

> > > > > > ### Comment · Reviewer_88d6 · 2023-11-22
> > > > > >
> > > > > > I acknowledge the motivation of making insignificant bits accurate under the prerequisite of not sacrificing the accuracy of significant bits. But I am not convinced of the prerequisite, i.e., the objective functions are conflicting. For simplicity, let A be the square loss, and B is the regularizer ($\lambda O_{reg}$). Authors claim that minimizing A+B will obtain a solution that is no worse than (under the criterion A) the solution of minimizing A. This is opposite to intuition and might occur when the algorithm of minimizing A is not well designed. If there is some special mechanism that supports authors' claim and refutes the intuition, I think such a mechanism is the central part of this paper and should be explained in more detail.

---

> ### Author Response · Authors · 2023-11-22
>
> Added clarification for non-conflicting nature of each objective component,
>
> Component 1 : $O_{pred}$ (MSE) - Nature: Non-negative.
> Component 2 : $O_{reg}$ (Cross Entropy) - Nature: Non-negative.
>
> Magnitude comparison in a general case, for relative assessment of prediction accuracy of the same quantity:
>
> $O_{reg}$ >= $O_{pred}$, Since, Cross Entropy generally greater that MSE for the same prediction assessment. Here, the authors agree with the reviewer that if the overall loss was $O_{pred}$ + $O_{reg}$, magnitude of one would overshadow the other.
>
> However, in the proposed setting the same is relatively weighted, that is overall loss = $O_{pred}$ + $\lambda$$O_{reg}$, where $\lambda$ is the hyper-parameter that is tuned to make sure the relative weighting is such that one does not overshadow the other. Note that in all ideal cases : $\lambda$ <= 1.
>
> The components thereby are non-conflicting since they have the same nature (one doesn't cancel the other) and they are relative weighted (one doesn't overshadow the other).

---

> ### Comment · Reviewer_88d6 · 2023-11-22
>
> Let's think the optimization in a mathematical way. For simplicity, we first define the following notations:
> 1. $O_{p}^{(1)}$ is the square loss of the solution by minimizing square loss (eq. 7).
> 2. $O_{r}^{(1)}$ is the cross entropy loss of the solution by minimizing square loss (eq. 7).
> 3. $O_{p}^{(2)}$ is the square loss of the solution by minimizing regularized loss (eq. 10).
> 4. $O_{r}^{(2)}$ is the cross entropy loss of the solution by minimizing regularized loss (eq. 10).
>
> In an ideal case, if both solutions are global minimizer, then it can be shown that $O_{p}^{(1)} \leqslant O_{p}^{(2)}$ and $O_{r}^{(1)} \geqslant O_{r}^{(2)}$. And when $N=1$, these two inequalities become equalities or strict inequalities at the same time. Then my question is what happens in experiments? Or what is expected to occur? Based on the relationf of $>$, $=$, and $<$ for these two pairs, there are 9 cases:
> 1. $O_{p}^{(1)} > O_{p}^{(2)}$ or $O_{r}^{(1)} < O_{r}^{(2)}$ (corresponds to 5 cases). This contradicts the intuition from theoretical results. This might happen in experiments since we cannot obtain the global minimizer. Then it would be necessary to explain the gap between theories and practices.
> 2. $O_{p}^{(1)} = O_{p}^{(2)}$ and $O_{r}^{(1)} = O_{r}^{(2)}$ (corresponds to 1 case). Then the proposed regularization is meaningless since it does not change the solution by minimizing (7).
> 3. $O_{p}^{(1)} = O_{p}^{(2)} , O_{r}^{(1)} > O_{r}^{(2)}$ or $O_{p}^{(1)} < O_{p}^{(2)} , O_{r}^{(1)} = O_{r}^{(2)}$ (corresponds to 2 cases). This cannot happen when $N=1$. If this is the case when $N>1$, the first case means that directly minimizing eq. (7) does not obtain a better solution than minimizing eq. (10), while the second case implies that directly minimizing eq. (10) does not achieve a better solution than minimizing eq. (7). This is contrary to intuition and needs more explanations.
> 4. $O_{p}^{(1)} < O_{p}^{(2)}$ and $O_{r}^{(1)} > O_{r}^{(2)}$ (corresponds to 1 case). This is the case that matches the theoretical intuition. If this happens, the prediction made by minimizing eq. (10) is less accurate, and the deviations from the solution will be larger as time increases.

---

> > ### Author Response · Authors · 2023-11-22
> >
> > Added clarification:
> >
> > The proposed setting is not performing individual optimisation of any one objective (Equation 7 or Equation 9). That is Equation~10 is not minimising cross entropy loss alone, but the relative weighted sum of squared loss and cross entropy.
> >
> > For further discussion please clarify the following:
> >
> > In the manuscript, $N$ is defined as the no of datapoints, what does it represent here ?

---

> > > ### Comment · Reviewer_88d6 · 2023-11-22
> > >
> > > Sorry for the typo in the last response, $O_{p}^{(2)}$ and $O_{r}^{(2)}$ are the square loss and cross entropy loss of the solution by minimizing $O_{total}$ (not cross entropy loss). The last response is asking the difference between the minimizer of square loss and that of the regularized loss (eq. 10). $N$ is the number of data points, consistent with the definition in the paper.

---

> ### Author Response · Authors · 2023-11-22
>
> Please note the discussion and intuition for the proposed objective:
>
> Starting with the fact that the no of datapoints is never equal to 1 in the proposed setting or the original LNN setting since LNN's are known to be notorious for their data requirements.
>
> Optimising squared error alone: $O^{(1)}_p$ will not guarantee convergence of Cross entropy objective, Since lower order bits would be poorly constrained and in stochastic optimisation settings achieving such precision would be not possible in finite time since the gradient magnitudes are small and unstable enough to be non contributing to any effective weight updates (true for most neural network based regression settings), and for the same $O^{(1)}_r$ which indicates the prediction accuracy of all bits of cannot be minimised beyond a point.
>
> Joint optimisation with relative weighted sum (proposed setting): Since it's difficult to exactly assess the convergence behaviour of joint optimisation setting, let's consider the following scenario: The LNN is optimised using squared error (Eq. 7) alone as the objective for some no of steps, and a state is achieved where the precision is no longer improvable because MSE is now unstable. Note that in such a stage the most significant bits would be predicted right. What one would require now is to introduce an objective that is more stable and directly contributing at such a stage, with similar magnitude and does not contradict what's optimised already. The scaled cross entropy term acts as the same, since it's conditioning each bit individually bit without relative weighting, the contribution from already optimised bits would be zero and hence has no adverse effect on the optimisation carried out by using squared error. The role of cross entropy at such as stage would be to improve the prediction accuracy of the lower order bits.

---

> ### Comment · Reviewer_88d6 · 2023-11-22
>
> Thanks for the detailed explanation.
>
> This response answers the relation between the second pair: $O_{r}^{(1)} > O_{r}^{(2)}$ because minimizing the square loss in practice will not lead to the convergence of lower order bits and the cross entropy loss will be large. This answer is consistent with theoretical intuition and can be a motivation for the proposed regularization.
>
> However, our purpose is to control the deviations from the true solution. Then the relationship of the first pair ($O_{p}^{(1)}$ and $O_{p}^{(2)}$) is also important. Furthermore, the regularized loss (eq. 10) gives lower order bits more significance (compared with the square loss). It is also important to discuss the influence of higher order bits and lower order bits on the deviations to motivate the regularization.

---

> ### Author Response · Authors · 2023-11-22
>
> An simplistic explanation would be by considering the scenario as in the previous comment:  The LNN is optimised using squared error (Eq. 7) alone as the objective for some no of steps, and a state is achieved where the precision is no longer improvable because MSE is now unstable. Note that in such a stage the most significant bits would be predicted right.
>
> Now the addition of cross-entropy would have the following implications: Since, larger order bits are already optimised the same won't contribute towards the Cross entropy value, that is their relative contribution would be zero, since they're already predicted right. Now the contributing part (lower order bits) are consequently optimised, the same would improve the accuracy of the overall solution, therefore improving the squared error too implicitly.
>
> On other note, an objective consisting of relative weighted MSE and cross-entropy is not something new in the deep learning literature, quoting specifically the class of Actor Critic methods from the Deep Reinforcement Learning (DRL) literature, the objective of such methods usually consists of two relatively weighted terms : the actor loss and the critic loss, where the actor loss is usually the cross entropy estimate based on the predicted and expected action spaces and the critic loss is a squared error between the predicted and expected state values, the applicability and relative superiority of actor critic algorithms on comparison with counterparts that use value loss or policy loss alone is well established.
>
> A classic and well established class of application of DRL is in the production of human level game playing agents, please find two attached references which uses a class of actor critic algorithms, and specifically uses an objective function similar to the proposed work : (sum of cross entropy and squared error);
>
> Silver, D., Hubert, T., Schrittwieser, J., Antonoglou, I., Lai, M., Guez, A., Lanctot, M., Sifre, L., Kumaran, D., Graepel, T. and Lillicrap, T., 2018. A general reinforcement learning algorithm that masters chess, shogi, and Go through self-play. Science, 362(6419), pp.1140-1144.
>
> Zhang, Hongming, and Tianyang Yu. "AlphaZero." Deep Reinforcement Learning: Fundamentals, Research and Applications (2020): 391-415.

---

> ### Comment · Reviewer_88d6 · 2023-11-22
>
> After the addition of cross entropy loss, we cannot guarantee that the larger order bits do not change. By minimizing the regularized loss, the loss of larger order bits may increase together with the loss decrease of lower order bits.
>
> Such a combinition is a kind of reweighting of the significance of different bits. It is important to clarify the reason for decreasing the significance of higher order bits by analyzing the influence of different bits on the deviations.

---

> > ### Author Response · Authors · 2023-11-22
> >
> > It's not specifically reweighing since the squared error is ever present during the optimisation process to provide baseline relative weighting. Also from prior literature such as the DRL literature quoted in the prior comment, the convergence of such an objective is well established and is not subjective of this work.

---

> > > ### Comment · Reviewer_88d6 · 2023-11-22
> > >
> > > Thanks for providing the related literatures. But my central concern is not the convergence. My concern is about the comparison between minimizing the square loss and minimizing the regularized loss. Currently, I am not convinced on the explanations about "minimizing the regularized loss will lead to a smaller deviation from the true solution" based on two facts:
> > > 1. Minimizing the regularized loss cannot guarantee more accurate higher order bits and lower order bits at the same time.
> > > 2. The relationship between deviations and different order bits is not discussed.

---

> ### Author Response · Authors · 2023-11-22
>
> The squared loss is analogous to the value loss in actor critic and the regularisation is analogous to the policy loss, since it can be seen that the same is assured to converge without conflicts producing one of the worlds best game playing agents, there is no reason to believe there would be any form of conflicts in the objective itself.
>
>
>
> Also it would be great to provide solid grounds for believing otherwise, or provide an example literature where such an objective loss is proven to diverge or conflict each other's optimality.
>
> 1. The base intuition is that minimising the regularisation component would guarantee that every predicted bit is exactly the expected bit. Added to the same the squared loss ensures that the higher order bits are predicted right anyway , hence the regularisation builds upon what's provided by squared error to condition explicitly on even the lowest bit of interest, thus improving the overall prediction. In short, it's quite obvious that jointly minimising the regularised loss with the squared loss component present improves the precision, on top of that the regularisation method as proposed here is unique for that fact that one can provide explicit conditioning on bits of significantly lower order magnitude, thus improving the overall precision of the proposed solution.
>
> 2. Solution deviation is maximum when the higher order bits are mis-predicted, however in chaotic systems the deviation would still show up even if bits of lower considerably lower order magnitude contribution is mis-predicted, when the solution is estimated for considerably large time-intervals, hence to obtain a valid solution space for a large enough time interval of interest, a minimum precision scale is necessary for each system. For which the proposed work introduces a scalable method by which arbitrary precision standards could be achieved.

---

> > ### Comment · Reviewer_88d6 · 2023-11-22
> >
> > The similarity between the proposed regularization and DRL is a good story but is not sufficient to support the regularization and the complex algorithm:
> > 1. The success of a method in one area does not promise its success in another area. AlphaZero is designed to play chess, shogi and go, while the task here is learning a chaotic dynamical system. These are two completely different problems. Even in game playing, AlphaZero only performs best on several games.
> > 2. A key idea in one model itself cannot make a new model success. The loss design is just one of many techniques to make AlphaZero work.
> > 3. To my understandings, the similarity is very limited. In the Q Actor Critic algorithm, the square loss is used to learn parameters of Q function, while the cross entropy loss is used to learn parameters of policy. The Q function and policy are two independent parts of the model. While in the proposed regularization, both losses are based on the same set of parameters.
> >
> > Finally, other reviewers also point out the problem about the algorithm and explanations. Reviewer HA5A says "I believe that the paper would greatly benefit from a simpler algorithm and better explanation" and asks for "some experiments showing that varying machine precision is not enough". Reviewer 2inG asks for "applying their method to more challenging datasets". Therefore, I think all reviewers are not convinced. Thus, it would be better to reorganize the paper, clearly demonstrate the motivation, explain the algorithm in detail, and conduct more convincing experiments and comparison.

---

> ### Author Response · Authors · 2023-11-22
>
> The understanding outlined is partially accurate except for the fact that for most of the modern actor critic implementations a single multi-output network is used to model both the Q or value function and the policy function, which is optimised using the relative weighted sum of both the value and policy losses. It's understandable that one algorithm cannot perform nominally in all conditions and there would be inherent limitations. The point we authors are trying to raise is that the components in the joint objective function is non-conflicting.
>
> The proposed work is a proof of concept for the relative scalability of the same in terms of achievable precision and the consequences the same in achieving more accurate long term solutions for systems that exhibit chaotic behaviour. More complex systems such as ones that require more complex constraints such as contact constraints are beyond the scope of the current study. however it maybe noted that the generality of the proposed approach extends beyond the augmentation of Vanilla-LNN.

---

> > ### Comment · Reviewer_88d6 · 2023-11-23
> >
> > After the long discussion, it becomes clearer why the regularization works. There are some further questions that I would like to confirm with authors.
> > 1. Is the LNN large enough to make the problem realizable (i.e., there exists an LNN with 0 loss)? If this is the case, then minimizers of the square loss and regularized loss are equivalent, which implies that the cross entropy loss does not hurt the square loss. The difference between these two losses are from practical optimization.
> > 2. Is there any noise in the training? In common machine learning tasks, labels may have noise. In this paper, the cross entropy loss forces LNN to have accurate output on lower bits, which may suffer from the risk of fitting noise. If the task here has no noise, then this is not an issue.
> > 3. Square loss alone cannot guarantee accurate prediction on lower bits, while square loss and cross entropy loss can. Then a natural question is: can cross entropy loss alone guarantee accurate prediction on lower bits? To my current understandings, the answer is yes. And I am wondering can we just minimize the cross entropy loss?

---

> > > ### Author Response · Authors · 2023-11-23
> > >
> > > 1. The optimization setting used in this study is the same as any other, which is by using Adam, a variant of the stochastic gradient descent strategy. It's important to note that all gradient descent algorithms regardless of how simple the estimation problem is (Ex. linear regression from perfectly noise free data), provides only asymptotic convergence guarantees. That is the true solution with zero loss can never be obtained in finite no of optimiser update steps. However, the convergence of the optimiser remains considerable for loss values higher than certain magnitude limits (problem specific). The same is the reason why the objective as proposed here is relevant, since squared error quickly diminishes when most of the most significant bits are predicted right, an extra regularisation (cross entropy) is required to maintain the a significant enough objective value and provide conditioning on the lower bits of interest.
> > >
> > > 2. The proposed method is used to estimate solutions of dynamic systems by implicitly modelling the Lagrangian, strictly based on the Lagrangian equation. Because of the same the approach does not require explicit data on the Lagrangian estimate (estimated directly), the actual Lagrangian is a true sample drawn from the Lagrangian equation of the dynamic system of interest. In short, the data used can be considered non-noisy as there is no source for the same.
> > >
> > > 3.  The intuition is right to assume that cross-entropy alone can guarantee accurate predictions, however it's worth noting again that the optimiser used is based on gradient descent which allows for asymptotic convergence guarantees only, the same would mean that in any given finite time (no of training steps) there would be a non-zero loss estimate. The same leads to the fact that a more elaborate constraint has to be used to provide constraint on quantities of interest wherever possible in order to improve the solution. One situation to note is that the cross entropy value for mis-predicting one bit from the set of bits of interest would be the same regardless of which order of magnitude the bit represents, as the same could happen at any instance of training the addition of the squared loss component stabilises the estimates formed in terms of their relative validity. In short it's better to loose the last bit (least order of magnitude) as compared to the first (highest order of magnitude).

---

> > > > ### Comment · Reviewer_88d6 · 2023-11-23
> > > >
> > > > Thanks for detailed explanations. The answers of 2 and 3 completely solve my questions, and I think the motivation of the regularization is convincing now. It would be better to contain part of the discussions in the paper to make the proposed method clearly motivated.

---

> > > > > ### Author Response · Authors · 2023-11-23
> > > > >
> > > > > The authors of this work thank the reviewer for his patience throughout this discussion and would add crucial and invaluable bits from the same as a extended discussion indicating the motivation of this work, since there is an explicit limit on the no of pages of the main body of the manuscript, the same would be added as a relevant appendix.

---

### Official Review · Reviewer_2inG · 2023-10-27

**Soundness:** 2 fair
**Presentation:** 2 fair
**Contribution:** 1 poor
**Rating:** 3
**Confidence:** 4

**Summary:**

This paper proposes a variant of the Lagrangian Neural Network (LNN) model for inducing higher precision outputs. The authors are motivated by chaotic systems, where slightly-inaccurate predictions can diverge quickly from the ground truth. In particular, the authors propose to output each binary bit of a traditional LNN's output, and they introduce several new regularization terms to supplement the regular LNN objective towards the goal of higher binary precision. The authors test their proposed model on the double pendulum and Henon-Heiles chaotic systems, with improvements over the original LNN architecture in the amount of steps before the predictions of each chaotic system's state diverges from the truth.

**Strengths:**

The method is unorthodox in the sense that neural network predictions are typically not performed as classification over the significant bits of the output. However, the authors design a training policy that carefully considers possible issues during training (such as the increasing $\mu_{TC}$ scale term and the proposed regularization term), which is appreciated.

For instance, $O_{reg}$ is used to supplement, not replace, the $O_{pred}$ MSE loss, and $O_{TC}$ is a good heuristic to deal with the situation in which the Lagrangian of a given system does not have a known analytical form. The increase weight of $\mu_{TC}$ is also a good solution to reduce instability from the initial transient stages of training.

The authors also compare between various choices of the precision parameter $k$, and they empirically show that the results of using low $k$ values are similar to the results of the original LNN model, which is an interesting result.

**Weaknesses:**

There is little to no explanation, intuition, or motivation about why this method should be superior to standard regression techniques optimizing with mean squared error. In general, there are several methodological concerns I have. For instance, computing the explicit Lagrangian using eq. 8 and comparing it with the ground truth MSE loss seems equivalent to the standard LNN formulation. Thus, the novelty of this method is in the regularization term $O_{reg}$ and in the regularization method to deal with unknown Lagrangians for the underlying system. Can the authors provide some intuition for why $O_{reg}$ is added and why it improves performance? Also, in what sense does $O_{reg}$ provide a regularization effect?

There is also no discussion and comparison to prior variants of LNNs. One such paper is Finzi et al., 2020, which also performs experiments on the double pendulum. I would strongly recommend the authors perform numerical experiments to compare against other prior LNN variants, not just the original LNN model. Furthermore, there is little to no discussion in the introduction about these LNN variants. At the very least, I would like to see some discussion about prior improvements to the LNN/Hamiltonian neural network (HNN) architectures.

Furthermore, the evaluation for this method seems a bit limited. Recent extensions of LNNs and HNNs target more difficult problems, such as 5-pendulums (Finzi et al., 2020) and pendulums with friction (Zhong et al., 2021). Given these prior works, I would also strongly recommend the authors add more challenging case studies (e.g., any of the ones mentioned earlier) and compare to prior methods.

In summary, if the authors wish to convince the readers of the novelty and contribution of their work, I would recommend adding a deeper explanation and intuition for this method, adding more difficult test cases, and adding comparisons with other LNN variants (not just the baseline paper).

**References:**
* Finzi, M., Wang, K. A., & Wilson, A. G. (2020). Simplifying hamiltonian and lagrangian neural networks via explicit constraints. Advances in neural information processing systems, 33, 13880-13889.
* Zhong, Y. D., Dey, B., & Chakraborty, A. (2021). Extending lagrangian and hamiltonian neural networks with differentiable contact models. Advances in Neural Information Processing Systems, 34, 21910-21922.

**Questions:**

* In $O_{reg}$, the least significant bits in the output appear to be weighted the same as the most significant bits. Is there a particular reason for this? Did the authors try a relative weighting between the most and least significant bits?
* In computation of $L_{pred}$, the authors mention that they round the sigmoid output of the model for each bit. How is this implemented in a differentiable way to allow for backpropagation through $O_{pred}$?

---

> ### Author Response · Authors · 2023-11-20
>
> Regarding weaknesses pointed out :
>
> The authors of this paper would like to thank the reviewer for the insights provided to improve the manuscript. The additional component (Eq. 9) in the objective function helps to explicitly condition on each significant bit of interest on the binary representation of the lagrangian estimate, thus improving the overall precision of the same. The same would enable the solution validity of deterministic chaotic systems to be extrapolated to larger domains of interest, or in other words the term helps in improving the generalizability of the solution space formed beyond what’s possible using alternative literature, which is demonstrated in Figs. 2(b) and 3(b), the same can also be confirmed by direct comparison with relative error margins obtained by (Finzi et al., 2020) on their version of experiments performed on the double pendulum system on similar time domains. Thus proving a regularization effect on the estimated solution space.
>
> A discussion outlining the relative comparison has been added quoting the previous works focused on improving the original LNN formulation and what sets the proposed approach apart from the same.
>
>
> The authors of this paper acknowledge that there are specific works that focus on improving the original Lagrangian Neural Network formulation, however this work takes a different approach to augment the original formulation. Specifically for estimating accurate long term solutions of dynamic systems that exhibit deterministic chaotic behavior. An additional section has been added in the conclusion section on the revised manuscript that specifically compares the error margins obtained as per the proposed approach and comparable experiments done in other LNN variants such as Finzi et al 2020, Zhong et al., 2021  which aims to improve accuracy and data efficiency by the addition of explicit  physical constraints. It’s also worth pointing out that the approach proposed in the work (the specific neural network architecture augmentations and the precision regularization) is general enough to be adapted to other formulations.
>
> Anders to specific Questions :
>
> An straightforward explanation would be that the relative squared error approximates heavily on solution space approximates where there is considerable deviation from the expected solution space. In other words the squared error approximates has a natural weighting towards the most significant bits of interest, due to the relative increase of its penalty magnitude in the event of misprediction. The regularization term acts as the explicit and equally weighted conditioning for estimating the relative accuracy of the predicted set of significant bits.
>
> The gradients are estimated by bypassing the rounding operation, although the rounding operation is important to obtain the Lagrangian estimate, the operation would cause the network to be rendered non-differentiable. In other words, we customize the gradient computation step such that  the effects of the rounding operation are non reflective on the gradient map.

---

> > ### Comment · Reviewer_2inG · 2023-11-20
> >
> > Thanks for your clarifications and answers to my questions.
> >
> > However, I am still not convinced about the contribution of this work in the context of prior LNN modifications. The authors mention that an additional section was added to the conclusion to compare against other LNN variants such as Finzi et al., 2020 and Zhong et al., 2021. Since this is a key point of mine, I would like to see this additional discussion explicitly. Would the authors be able to quote the passage that they added?
> >
> > Also, a suggestion I still have (from my initial review) for the authors is applying their method to more challenging datasets, as in Finzi et al., 2020 and Zhong et al., 2021. I'm afraid that without a comparison with prior/baseline methods on more challenging settings, it is difficult to see how this work fits into the context of prior works.
> >
> > As such, I will maintain my score as it is for now. However, I would be willing to re-evaluate my score if new quantitative results and comparisons were to be presented.

---

> > > ### Author Response · Authors · 2023-11-23
> > >
> > > To better motivate the work proposed here, a new section : Appendix D has been added that relatively compares the strengths and weaknesses of the proposed work with the existing literature.

---

> > > > ### Comment · Reviewer_2inG · 2023-11-23
> > > >
> > > > I appreciate the expansion of the manuscript to include a comparison of the strengths and weaknesses of the proposed work with prior works. Would the authors be able to quote any passages relevant to our discussion here?

---

> > > > > ### Author Response · Authors · 2023-11-23
> > > > >
> > > > > Specifically: Appendix section D.

---

### Official Review · Reviewer_HA5A · 2023-10-27

**Soundness:** 3 good
**Presentation:** 2 fair
**Contribution:** 1 poor
**Rating:** 3
**Confidence:** 3

**Summary:**

Lagrangian neural networks have emerged as a promising approach to learning the dynamical behavior of a system from data. However, its limited precision hurts the prediction of long-time sequences, in particular, if the system is chaotic. The paper introduces a modification of the LNN framework where precision is explicitly modeled and shows that it improves prediction error in two empirical settings.

**Strengths:**

The problem studied is well justified. The description of the neural architecture used, and of the task studied is clear.

**Weaknesses:**

I found the algorithm extremely convoluted, which is, in general, not a good sign for its robustness. It could be justified if simpler solutions, such as changing the default float32 in Jax to float64, do not work. However, the authors do not provide any data points suggesting that these simpler approaches do not work.

The experiments are too limited for me to be able to judge if the approach works.

Additionally, I found it a bit weird that only the original Lagrangian Neural Network paper is cited (and compared to), as papers improving the idea have been published since then (e.g., Finzi et al 2020).

**Questions:**

See Weaknesses.

---

> ### Author Response · Authors · 2023-11-20
>
> The authors of the paper would like to thank the reviewer for his considerate set of comments and a revised manuscript is created to address the crucial points. While the precision can be increased from 32 to 64, it should be noted that errors in integration will creep into the solution, which for a chaotic system such as those studied in this work, will result in significant deviations from the true solution as time increases. The key idea in this work is to train the network to understand a chaotic system for a certain amount of time, and then use the trained network to extrapolate to much larger times.  One essential point to note is that the proposed approach enables adding an arbitrary level constraint on the precision scale of the solution space of any dynamic system of interest. The novelty of the work is solely based on the explicit conditioning mechanism outlined, which allows arbitrary constraints on each significant digit of interest. Varying the machine precision from float32 to float64 allows for the accommodation of an additional significant digits, however the same would have minimal effect on the overall precision of the solution, since in finite time stochastic optimization settings such the parametric optimization of a neural network the impact of the lower order digits on the relative error term is negligible (by virtue of its magnitude), the same consequently mean a negligible contribution or effect on each gradient update operation.
>
>
> The proposed approach is a proof of concept, where explicit precision constraints could be enforced on the solution space of any given dynamic system, and as an example application, we demonstrate the applicability of the approach on dynamic systems that exhibit deterministic chaos. The cases chosen are two well known chaotic systems, i.e., the double pendulum and the henon-heiles problems. As seen in Figs. 2(b) and 3(b), the network is able to predict the solution up to 20 and 60 seconds, respectively. Furthermore, the parameter ‘k’ which decides the number of output branches of the neural network (the number of significant digits of interest) can control the precision required by the user for the problem at hand. Therefore, we feel that the present studies show the efficacy of the proposed formulation. The fact that precision can be enforced to a required level in the solution space possible using the proposed approach is the main novelty of the present work. Indicative results outlining the relative change in the error margin upon the variation of the scale parameter ‘k’ is outlined in figures 2(b) and 3(b) with an observable consistency in the trends shown.
>
> The authors of this paper acknowledge that there are specific works that focus on improving the original Lagrangian Neural Network formulation. An additional section has been added in the conclusion section on the revised manuscript that specifically compares the error margins obtained as per the proposed approach and comparable experiments done in other LNN variants such as Finzi et al 2020, Zhong et al., 2021  which aims to improve accuracy and data efficiency by the addition of explicit physical constraints, the same improves the relative accuracy in the time domain of interest, and not extrapolate the solution beyond the same.. However, the present work takes a different approach to augment the original formulation. Specifically, for estimating accurate long term solutions of dynamic systems that exhibit deterministic chaotic behavior. It’s also worth pointing out that the approach proposed in the work (the specific neural network architecture augmentations and the precision regularization) is general enough to be adapted to other formulations.

---

> > ### Comment · Reviewer_HA5A · 2023-11-20
> >
> > I would like to thank the authors for the clarifications they provided.
> >
> > Unfortunately, I still think the paper needs substantial modifications before being in a good enough state to be accepted.
> > As touched upon in my review, here are the points I think should be covered in an updated version:
> > - some experiments showing that varying machine precision is not enough (on top of the discussion that the authors provided in their answer).
> > - as also mentioned by other reviewers, I believe that the paper would greatly benefit from a simpler algorithm and better explanation.
> > - a better comparison to previous works would help in understanding the specific contributions of this paper.

---

> > > ### Author Response · Authors · 2023-11-23
> > >
> > > To better motivate the work proposed here, a new section : Appendix D has been added that relatively compares the strengths and weaknesses of the proposed work with the existing literature.

---

### Meta-Review · Area_Chair_3z8w · 2023-12-06

**Metareview:**

This paper studies the application of Lagrangian Neural Nets (a type of physics-informed model that learns Lagrangians of dynamical systems) for chaotic systems. In such systems, the precision of the model is crucial for simulation quality, so the paper proposes a new loss to enforce high precision.

There was an unusually long discussion between the authors and some reviewers. Despite this intensive process, the reviewers still broadly agree that the paper is not yet ready for publication, mostly because the motivation and the experiments are not convincing. In fact this is reinforced precisely by the fact that even after a very long discussion, one of the reviewers is still unclear about how and why the proposed new loss function should be an improvement.

I understand that the authors want to use very opportunity to improve their chances of acceptance, and that openreview does not pose a limit on such communication. But forcing a long and detailed discussion does not automatically imply that a paper has to be accepted, or that reviewers are obliged to change their mind. All reviewers stand by their negative evaluation, and I thus can not recommend the paper for ICLR.

**Justification For Why Not Higher Score:**

Broad agreement among reviewers.

**Justification For Why Not Lower Score:**

N/A

---

### Decision · Program_Chairs · 2024-01-16

Reject